# A Cartilage Matrix Protein Regulates Collagen Synthesis in Mantle of *Magallana gigas* (*Crassostrea gigas*) under Ocean Acidification

Ting Zhu [1,2,3], Chang Liu [1,2,3,*], Zhaoqun Liu [1,2,3], Yuqian Gao [1,2,3], Xiaoyu Xin [1,2,3], Lingling Wang [1,2,3,4,*] and Linsheng Song [1,2,3,4]

1   Liaoning Key Laboratory of Marine Animal Immunology, Dalian Ocean University, Dalian 116023, China; zhuting199@hotmail.com (T.Z.); liuzhaoqun@hainanu.edu.cn (Z.L.); gaoyuqian105@hotmail.com (Y.G.); xinxiaoyu83@hotmail.com (X.X.); lshsong@dlou.edu.cn (L.S.)
2   Dalian Key Laboratory of Aquatic Animal Disease Prevention and Control, Dalian Ocean University, Dalian 116023, China
3   Liaoning Key Laboratory of Marine Animal Immunology and Disease Control, Dalian Ocean University, Dalian 116023, China
4   Functional Laboratory of Marine Fisheries Science and Food Production Processes, Qingdao National Laboratory for Marine Science and Technology, Qingdao 266235, China
*   Correspondence: liuchang@dlou.edu.cn (C.L.); wanglingling@dlou.edu.cn (L.W.)

**Abstract:** The shell biosynthesis of oysters plays a critical role in protection against environmental stress, in which cartilage matrix proteins (CMPs) determine the mineralogical and crystallographic properties of the shell. In the present study, a cartilage matrix protein (designated as *Mg*CMP1) was identified from the Pacific oyster *Magallana gigas* (*Crassostrea gigas*) with the objective of understanding its possible role in shell formation. The open reading frame (ORF) of *Mg*CMP1 was 1815 bp, encoding a polypeptide of 605 amino acids with two von Willebrand factor (VWA) domains. The mRNA transcript of *Mg*CMP1 was expressed constitutively in all examined tissues with a higher level in the mantle, especially highest in the middle fold (MF) of the three folds of the mantle. In addition, the interaction between recombinant protein MgCMP1 (rMgCMP1) and recombinant protein bone morphogenesis protein 7 (rMgBMP7) was identified in vitro. After injection of dsRNA to inhibit the expression of *Mg*CMP1, the mRNA expression level of *Mg*collagen I and *Mg*collagen X in the MF of the mantle significantly decreased. After pre-puncturing and acidification treatment (pH 7.8), the thickness and length of the new formation shells were lower than those in control group (pH 8.1), and the positive hybridization signals of the *Mg*CMP1 mRNA transcript in the three mantle folds were obviously weakened, especially in the MF, whereas the mRNA expression level of *Mg*CMP1, *Mg*collagen I and *Mg*collagen X in the MF of mantle decreased significantly. These results suggested that *Mg*CMP1 was involved in regulating the expression of *Mg*collagen I and *Mg*collagen X in the MF of the mantle in response to ocean acidification (OA).

**Keywords:** ocean acidification; *Magallana gigas* (*Crassostrea gigas*); cartilage matrix protein; middle fold of the mantle; collagen

**Key Contribution:** The mRNA of *Mg*CMP1 was highly expressed in the MF of the mantle. OA severely inhibited the growth of oyster calcified shell and the expression of *Mg*CMP1, *Mg*collagen I and *Mg*collagen X. The activation of the CMP1-BMP7 pathway was inhibited under acidification, which further affected *Mg*collagen I and *Mg*collagen X synthesis as well as the shell growth of the oysters.



## 1. Introduction

OA is a global marine ecological disaster, which probably brings about reduction in biodiversity and a radical alteration in marine ecosystems [1,2]. Previous studies have

shown that the ocean pH is expected to decrease from 8.1 to 7.8 by the end of 2100 [3,4]. Bivalve shells consist of more than 95% calcium carbonate minerals and a small amount of organic matrix [5–7]. OA can reduce the saturation state of calcium carbonate ($CaCO_3$) and affect the calcification rate of shells [8–10]. The calcification rates of *Mytilus edulisand* and *M. gigas* decrease by 25% and 10%, respectively, at $CO_2$ concentrations of 740 ppm [9]. The mantle is the main tissue for shell formation [11]. Different regions and folds of the mantle are involved in the formation of different shells. The periosteum layer and prismatic layer are formed by secretions from the marginal zone [11–13], and the mantle marginal zone can be further divided into the inner fold (IF), MF and outer fold (OF) of the mantle [14]. OA inhibits the secretion of organic matrix in the MF of the mantle to inhibit the formation of shells [15–17]. With the deepening understanding of the mechanisms of shell formation affected by OA, studies have gradually extended from phenomenon observation to the molecular level. Recent studies have found that a new family of molecules, cartilage matrix protein (CMP), may play an important role in shell formation in shellfish [18].

CMP (also known as matrilin-1) belongs to the matrilin family and is a type of extracellular matrix protein that was first isolated and identified in 1992 [19]. There are four currently known matrix proteins in vertebrates: matrilin-1, matrilin-2, matrilin-3 and matrilin-4. Similar to other matrilins, CMPs share a structure consisting of two tandem von Willebrand factor domains (VWA domain), an epidermal growth factor-like domain (EGF domain) and a coiled coil α-helical module [20]. Analysis of the evolutionary relationships revealed that the VWA modules of matrilin-2, matrilin-3 and matrilin-4 were possible replicated from the second VWA module of matrilin-1 [21]. The matrix proteins interact with other extracellular matrix proteins via the metal ion-dependent adhesion site (MIDAS) of the VWA domain [22,23]. CMP is mainly expressed in the cartilage of vertebrates, and it plays important physiological roles in the development of the osteoarticular system and in maintaining the homeostasis of various connective tissue structures, mainly via interacting with bone morphogenesis protein (BMP) and modulating collagen [24]. CMP may serve as an "instructive matrix" component, binding to bone morphogenesis proteins 2 (BMP2), 4 (BMP4) and 7 (BMP7), to participate in the regulation of corresponding downstream signaling pathways [25,26]. CMPs act as modulators of collagen fibrillogenesis in cartilage, which may affect the organization and integrity of the human cartilage growth matrix [27–30]. Mice that lack CMP and collagen IX exhibit ultrastructural abnormalities in the shortened long bones [27,31]. The knockdown of CMP in zebrafish results in growth defects, disturbed craniofacial cartilage formation and decreased collagen II deposition [32].

In mollusks, an organic matrix provides the framework for the deposition of $CaCO_3$ crystals [6,23,33]. CMPs and VWA domain-containing proteins have also been identified in mollusks and play an important role in shell formation. The CMPs in oyster *Crassostrea virginica* and conch *Biomphalaria glabrata* harbor the typical VWA domains, missing the EGF-like domain and the coiled coil α-helical module, which are different from the matrilin family members in vertebrates [34,35]. Currently, research on CMPs and VWA domain containing proteins in invertebrates is mainly focused on their expression pattern in tissues and their possible role in regulating the nacreous shell layer. For instance, in *Pinctada fucata*, CMP was found to be highly expressed in the mantle. Aragonite-binding protein (Pif 97) containing one VWA domain was mainly expressed in the nacreous layer of *M. gigas* [23]. Blue mussel shell protein (BMSP) with four VWA domains was involved in maintaining nacreous matrix formation in *Mytilus galloprovincialis* [36].

The Pacific oyster *M. gigas* is an important cultured species worldwide, and a model organism for investigation mechanisms of adaptation to the marine environment [37]. OA causes the dissolution of adult oyster shells, leading to significant ecological and economic losses. It has been demonstrated that OA could inhibit shell formation by affecting the secretion of chitin synthase (chitin) and carbonic anhydrase (CA) in *M. gigas* [38,39]. CMP may be involved in shell formation in oysters; however, whether OA affects shell formation by regulating CMP is unclear and needs to be further studied. In this study, the regulation

mechanism of *Mg*CMP1 in shell formation under acidification treatment will be explored; this could contribute to better understanding how OA affects marine calcifiers.

## 2. Materials and Methods

### 2.1. Oyster Treatment, and Sample Collection

2.1.1. Obtaining and Maintenance of Oysters

The animal experiments in this study are in accordance with the animal ethics guidelines approved by the Ethics Committee of Dalian Ocean University. The Pacific oyster *M. gigas* (2–3 years old) was purchased from a farm (Dalian, China). In this study, 20–30 oysters were acclimated in a tank, whose size was 200 cm × 100 cm × 50 cm, with filtered and aerated seawater at 15–20 °C for a week. During the experiment, the oysters were fed with spirulina algae powder, and 2/3 of the volume of seawater was changed daily.

2.1.2. Aperture and Acidification Treatment

Eighty-one oysters were averagely and randomly divided into three groups, blank group, control group and OA group. Twenty-seven oysters in the blank group (pH 8.1 ± 0.05) were cultured for 6 weeks in aerated seawater. Twenty-seven oysters in the control group (pH 8.1 ± 0.05) underwent lateral pre-puncture in the closed side of the oyster shell adjacent to the adductor muscle (aperture treatment), and were cultured for 6 weeks in aerated seawater. Twenty-seven oysters in the OA group (pH 7.8 ± 0.05) under aperture treatment, were cultured for 6 weeks in aerated seawater with an air–$CO_2$ mixture (acidification treatment). The pH value was automatically controlled using an acidometer and pH probe (AiKB, Qingdao, China), and the pH value parameter was set to 7.8 ± 0.05 [39].

2.1.3. Measurement of Length and Thickness of New-Formation Shells

Nine oysters from the control and OA groups were numbered. At the 2nd, 4th and 6th weeks, the length and thickness of newly formed shells of the same oysters in the control and OA groups were measured using vernier calipers (Syntek, Huzhou, China) (Figure S1).

2.1.4. Sample Collection

The mantle marginal zone, separated into inner/middle/outer fold (IF/MF/OF), was collected from each group at 2, 4 and 6 weeks. Samples from the edge area of the mantle were added to in situ hybridization fixative (Servicebio, Wuhan, China) for subsequent in situ hybridization (ISH) experiments. To detect the distribution pattern of *Mg*CMP1 in normal oyster tissues, the gill, hepatopancreas, mantle marginal zone, adductor muscle, haemocytes and labial palp of nine oysters were collected without any treatment after culturing for a week. Tissues from three oysters were pooled together as one sample, and there were three replicates for each tissue (N = 3). Each sample was added into 1 mL TRIzol (Thermo Fisher Scientific, Waltham, MA, USA) reagent for RNA extraction.

### 2.2. Total RNA Extraction and cDNA Synthesis

A 1 mL volume of TRIzol and 0.2 mL of RNA Extraction Agent (Trans Gen Biotech, Beijing, China) were added into every 50–100 mg tissue for homogenization. After centrifugation at 10,000× *g* at 4 °C for 15 min, the colorless upper aqueous phase was taken for total RNA extraction [40,41]. The first chain cDNA was synthesized using total RNA as template and oligo dT-adaptor as primers. The synthesis reaction was performed at 42 °C for 30 min and terminated by heating at 84 °C for 30 s. The synthesized cDNA products were diluted to 1:20 for gene cloning and quantitative real-time PCR (qRT-PCR) analysis, and stored at −80 °C.

### 2.3. Gene Cloning and Sequence Analysis

By screening the genome of the oyster *M. gigas*, a gene encoding CMP was identified from National Center for Biotechnology Information (NCBI accession no LOC 105349064).

The ORF of *Mg*CMP1 was cloned from the cDNA library of the mantle by using specific primers (*Mg*CMP1-KL-F and *Mg*CMP1-KL-R, Table 1) and Pro Taq DNA Polymerase kit (Accurate Biology, China). The polymerase chain reaction (PCR) system consisted of Pro Taq DNA polymerase 0.25 µL, 10 × Pro Taq PCR buffer Ver 5 µL, MgCl$_2$ Solution 2 µL, dNTP Mix 1 µL, Template (1 µg/µL) 1 µL, *Mg*CMP1-KL-F Primer 5 µL, *Mg*CMP1-KL-R Primer 5 µL and DEPC water up to 50 µL.

**Table 1.** Sequences of the primers used in this study.

| Primer | Sequence (5′–3′) |
| --- | --- |
| Clone primers | |
| *Mg*CMP1-KL-F | TGATGAACGACCCGCTTA |
| *Mg*CMP1-KL-R | AAAATGTAGGCACGGCTGT |
| *Mg*CMP1-M-F | CCGGAATTCATGCTGACTTTCTTAGTTTTGTG |
| *Mg*CMP1-M-R | CCGCTCGAGATCAATATCAAAACACAGCTCGT |
| RT-PCR primers | |
| *Mg*CMP1-RT-F | ATCGTGAGTGCGTTCGACAT |
| *Mg*CMP1-RT-R | CACGTGACAGTCCATCCGTT |
| *Mg*EF-RT-F | AGTCACCAAGGCTGCACAGAAAG |
| *Mg*EF-RT-R | TCCGACGTATTTCTTTGCGATGT |
| *Mg*collagen X-RT-Fi | CGACACCGTGGTGACCAATA |
| *Mg*collagen X-RT-Ri | GCATTGCGCACTAACCTCAC |
| *Mg*collagen I-RT-Fi | ACCTCCAGGACCTTCGTTTG |
| *Mg*collagen I-RT-Ri | TATCCTTGCCGCTGGTGAC |
| *Mg*CMP1-RT-Fi | CCCGGTCGATCTTGTGTTCA |
| *Mg*CMP1-RT-Ri | ACTCCAATCGTGGCTCATCG |
| WEISH primer | |
| *Mg*CMP1-WEISH-F | CTGGGTATCTTTGTGCTTGC |
| *Mg*CMP1-WEISH-R | AAATCCAGTGTCGGTGCC |
| RNAi primer | |
| *Mg*CMP1-RNAi-F1 | GATCACTAATACGACTCACTATAGGGGCTCGCTCTAATGTTGCC |
| *Mg*CMP1-RNAi-R1 | GATCACTAATACGACTCACTATAGGGGGTTGAAAGGGAAAGTCGC |
| EGFP-Fi | GGATCCTAATACGACTCACTATAGGGATCCGACGTAAACGGCCACAAGT |
| EGFP-Ri | GGATCCTAATACGACTCACTATAGGGATCCTTGTACAGCTCGTCCATGC |

The BLAST algorithm was used for amino acid sequence analysis. The theoretical isoelectric point (PI) and molecular weight of *Mg*CMP1 were predicted using ExPasy's pI/Mw tool. The domain of *Mg*CMP1 was predicted by Simple Modular Architecture Research Tool (SMART). Based on the CMP amino acid sequences of different species, a phylogenetic NJ tree was constructed using MEGA 7.0 software (version no 1.0.0.0) to analyze the evolutionary relationships of *Mg*CMP1. Multiple sequence alignment analysis of *Mg*CMP1 was performed by the DNAMAN 9.0 program [42,43].

*2.4. qRT-PCR Analysis of mRNA Expression*

The qRT-PCR were performed using the Trans-Start Top Green qPCR Super Mix (Trans Gen Biotech, Beijing, China) reagent system (SYBR Green Master Mix 5 µL, ROX 0.2 µL, Forward primer 0.2 µL, Reverse primer 0.2 µL, cDNA template (1µg/µL) 2µL, DEPC water 2.4 µL, total volume 10 µL.) and on Quan Studio 6 Flex (Thermo Fisher, Waltham, MA, USA) reaction procedure (95 °C for 30 s. 95 °C, 5 s; 60 °C, 31 s, 40 cycles) [44]. Using the elongation factor gene (LOC105338957, *Mg*EF) from *M. gigas* in Table 1 as internal control, the relative expression level of *Mg*CMP1, *Mg*collagen I (LOC105326709) and *Mg*collagen X (LOC105346696) in Table 1 were analyzed using the comparative Ct method ($2^{-\Delta\Delta Ct}$ method) [45].

*2.5. Recombinant Expression and Purification of MgCMP1*

The sequence of the VWA domain in *Mg*CMP1 was amplified using primers *Mg*CMP1-M-F and *Mg*CMP1-M-R (Table 1). The *Mg*CMP1 PCR products with *EcoR* I and *Xho* I sites were cloned into expression vector pET-30a (+) and transferred into (DE3) *Escherichia coli* (*E. coli*) competent cells [46]. Positive transformants were screened with the primers *Mg*CMP1-M-F and *Mg*CMP1-M-R (Table 1), and incubated in LB liquid medium at 37 °C. The expression of rMgCMP1 was induced by adding isopropyl-β-dthiogalactoside (IPTG, final concentration 0.5 mM).

The denatured rMgCMP1 was purified using an Ni-NTA Sepharose column and pooled by elution with 400 mmol/L imidazole. The protein activity of rMgCMP1 was restored by dialysis using urea with concentration gradients (6, 5, 4, 3, 2, 1 and 0 M) and PBS buffer (three times) at pH 8.0, 4 °C, 12 h apart. RMgCMP1 protein was detected and quantified using 12% SDS-polyacrylamide gel electrophoresis (SDS-PAGE) and BCA methods and stored at −80 °C [46].

*2.6. The Molecular Interaction between rMgCMP1 and rMgBMP7*

The rMgCMP1 was prepared at different concentrations (2.22 μM, 1.66 μM, 1.25 μM, 0.94 μM and 0.71 μM), and rMgBMP7 was biotinylated with a Biotinylation Kit (GEN-EMORE, Shanghai, China). Based on previous studies, the in vitro interaction of rMgCMP1 and rMgBMP7 was detected using Bio-Layer Interferometry (BLI) and Octet K2 (Forte-Bio, Silicon Valley, CA, USA). The Octet K2 program was running according to Baseline, Loading, Association, and Dissociation [46]. The binding constant ($K_{on}$) and dissociation constant ($K_{off}$) were analyzed by ForteBio software, and the affinity ($K_D$) was obtained by fitting calculation [47].

*2.7. In Vivo RNA Interference (RNAi) Treatment of Oysters*

The cDNA fragments of *Mg*CMP1 and EGFP were amplified as templates to synthesize dsRNA using the primers *Mg*CMP1-Fi, *Mg*CMP1-Ri, EGFP-Fi and EGFP-Ri (Table 1). The dsRNAs were synthesized using Transcription T7 Kit (NEB, Ipswich, MA, USA).

Thirty-six oysters were randomly divided into four groups: blank group (without any treatment), seawater group (SW group, injections SW), EGFP-RNAi group (injections dsEGFP) and CMP1-RNAi group (injections ds*Mg*CMP1), with nine individuals in each group. To enhance the effect of RNAi, dsEGFP and ds*Mg*CMP1 (100 μL, 1 μg μL-1) were injected twice, 12 h apart. Twenty-four hours after the second injection, the MF of the mantle (N = 3) of each group was collected for total RNA extraction and cDNA synthesis. *Mg*CMP1-RT-Fi and *Mg*CMP1-RT-Ri primers (Table 1) and qRT-PCR were used to detect and evaluate the effect of RNAi.

*2.8. In Situ Hybridization*

2.8.1. RNA Probe Preparation

The fragments were cloned from the mantle cDNA library using primers *Mg*CMP1-WEISH-F and *Mg*CMP1-WEISH-R (Table 1). The fragments were purified and connected to the pGEM-T vector (Promega, Madison, WI, USA), then transferred into (DE3) *E. coli* competent cells [46]. Positive transformants were screened with the primers *Mg*CMP1-WEISH-F and *Mg*CMP1-WEISH-R (Table 1), and incubated in LB liquid medium at 37 °C. The recombinant plasmid *Mg*CMP1-pGEM-T was digested using *Nco* I or *Nde* I endonuclease The probe was synthesized using T7/SP6 RNA polymerase and DIG RNA labeling kit (Roche, Mannheim, Germany), and tested using a nucleic acid detection kit (Roche, Mannheim, Germany) [48].

2.8.2. ISH Techniques

Tissue sections were obtained from the edge area of the *M. gigas* mantle according to previous reports [49]. Firstly, the paraffin wax in the mantle tissue sections was replaced with 50% ethanol (xylene, 100%–90%–70%–50% ethanol), then fixed with 4% PFA, and di-

gested with protease K solution. Secondly, the mantle tissue sections were hybridized with digoxin-labeled RNA probes at 53 °C. Finally, the water in the mantle tissue sections was replaced with xylene (50%–70%–90%–100% ethanol, xylene) to facilitate color observation. ISH techniques are detailed in the previous report [50,51].

### 2.9. Statistical Analysis

All data are given as means $\pm$ S.D. Significant difference was determined by two-tailed Student's t-test, or by one-way analysis of variance (ANOVA) followed by Duncan multiple comparisons. Different letters indicated statistically significant difference at $p < 0.05$.

## 3. Results

### 3.1. The Sequence Characteristics and Phylogenetic Relationship of MgCMP1

The ORF of *Mg*CMP1 was of 1815 bp encoding a polypeptide of 605 amino acids (Figure S2A) with a predicted molecular mass of 67.28 kDa. There were two VWA domains (Figure S2B) in the amino acid sequence of *Mg*CMP1, which were located at residues 26–208 and residues 239–420. The theoretical isoelectric point (PI) of *Mg*CMP1 was predicted to be 4.71 using the computational pI/Mw tool of ExPasy.

The phylogenetic tree showed that these CMPs from vertebrates and invertebrates were separated into two major branches. *Mg*CMP1 was closely clustered with the CMP of *C. virginica*. (Figure S2C).

The amino acid sequences of CMPs were compared and analyzed, and it was found that there was high identity between *Mg*CMP1 and CMPs form other species, such as 31% identity with that of *Homo sapiens*, 32.98% identity with that of *Mus musculus*, 29.62% identity with that of *Danio Rerio*, 21.3% identity with that of *Fulmarus glacialis*, 31.6% identity with that of *Patella vulgate* and 74.79% identity with that of *C. virginica* (Figure S2D).

### 3.2. The Distribution of MgCMP1 mRNA in Different Tissues

The mRNA expression levels of *Mg*CMP1 in different tissues (hepatopancreas, labial palp, mantle, gills, adductor muscle and hemocytes.) of oyster were detected by qRT-PCR. The transcripts of *Mg*CMP1 were expressed in all tissues, and the expression level in mantle was the highest, being 8.67-fold ($p < 0.01$) higher than that in labial palp. *Mg*CMP1 mRNA was also highly expressed in hemocytes and gill, being 4.84-fold ($p < 0.01$) and 3.63-fold ($p < 0.01$) of that in labial palp, respectively (Figure 1A). There were no significant differences among adductor muscle, labial palp and hepatopancreas. Expression levels of *Mg*CMP1 mRNA in different folds of oyster mantles, including IF, MF and OF, were also detected. The expression level of *Mg*CMP1 in MF was the highest, being 1.43-fold and 2.98-fold higher than that in IF and OF ($p < 0.05$, Figure 1B).

### 3.3. Recombinant Protein of rMgCMP1 and Its Interaction with rMgBMP7 In Vitro

After IPTG induction, the purified rMgCMP1 was analyzed with 12% SDS-PAGE. A distinct band consistent with the predicted molecular mass of rMgCMP1 was observed, at about 73.9 kDa (Figure 2A).

The interaction between rMgCMP1 and rMgBMP7 in vitro was detected by BLI. At lower concentrations, no valid response of MgBMP7 was detected. With an increase in rMgCMP1concentration (from 0.71 μM to 2.22 μM), the binding affinity of rMgCMP1 to rMgBMP7 was enhanced, and the affinity $K_D$ was $5985 \times 10^{-8}$ mol$\cdot$L$^{-1}$ (Figure 2B). This indicated that rMgCMP1 and rMgBMP7 had specific binding activity.

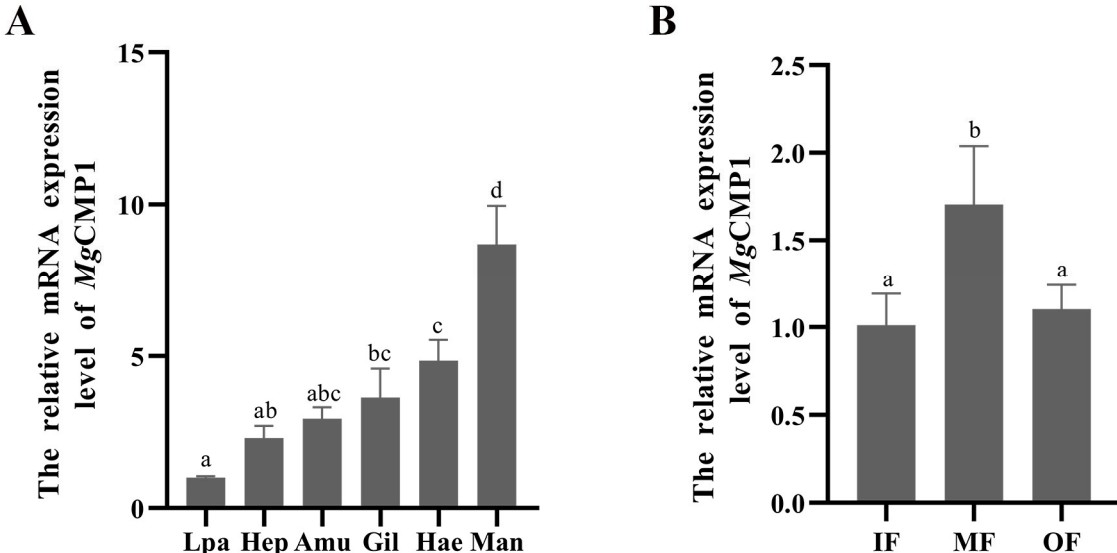

**Figure 1.** The expression level of *Mg*CMP1 mRNA in different tissues. (**A**) Gil, gill; Man, mantle marginal zone; Hep, hepatopancreas; Amu, adductor muscle; Lpa, labial palp; Hae, haemocytes. (**B**) IF, inner fold of the mantle; MF, middle fold of the mantle; OF, outer fold of the mantle. The different letters (a, b, c, and d) indicate significant differences ($p < 0.05$). Each value is shown as mean ± S.D. (N = 3).

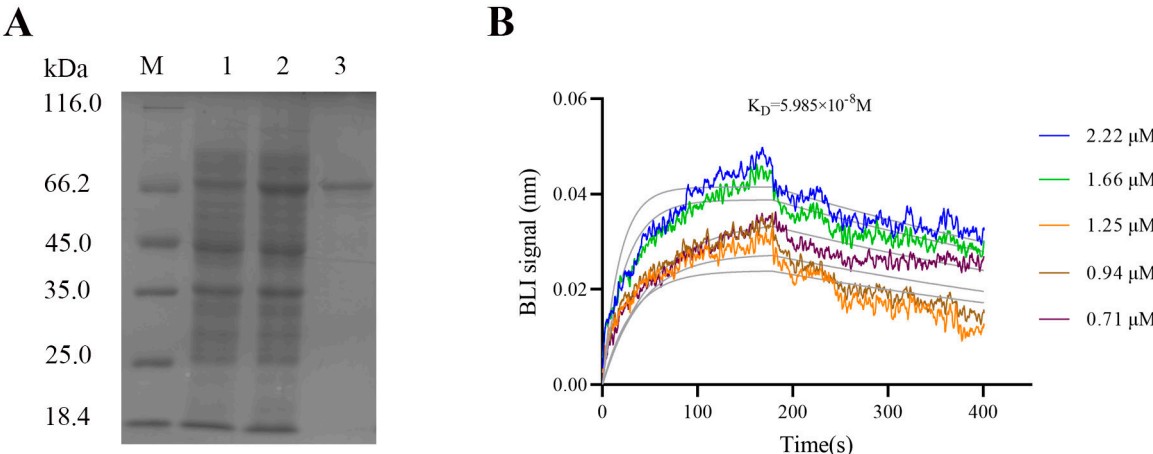

**Figure 2.** The recombinant protein of rMgCMP1 and its interaction with rMgBMP7 in vitro. (**A**) SDS-PAGE analysis of the recombinant protein rMgCMP1. Lane M, standard protein molecular weight marker; Lane 1, negative control (without IPTG induction); Lane 2, induced recombinant protein rMgCMP1; Lane 3, purified recombinant protein rMgCMP1. (**B**) The binding capability of rMgCMP1 with rMgBMP7 measured by BLI assay.

### 3.4. The mRNA Transcripts of Mgcollagen I and Mgcollagen X in MgCMP1-RNAi Oysters

In order to explore the relationship between CMP and collagen, the expression levels of *Mg*collagen I and *Mg*collagen X in the MF of mantle were examined after *Mg*CMP1 had been inhibited by *Mg*CMP1-dsRNA. The expression level of *Mg*CMP1 decreased significantly (0.49-fold of that in EGFP-RNAi oysters, $p < 0.01$) at 24 h after the injection of *Mg*CMP1 dsRNA (Figure 3A). Meanwhile, the expression levels of *Mg*collagen I and *Mg*collagen X were also significantly decreased, being 0.21-fold ($p < 0.001$) and 0.35-fold ($p < 0.05$) of those of EGFP-RNAi oysters (Figure 3B,C) at 24 h after the injection of *Mg*CMP1 dsRNA.

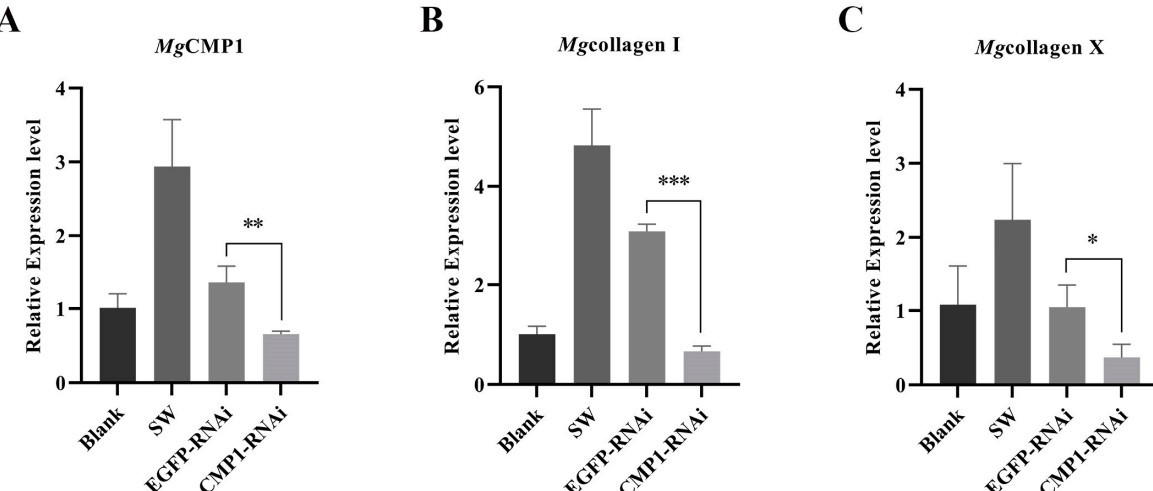

**Figure 3.** The mRNA expression levels of *Mg*collagen I and *Mg*collagen X in the MF of mantle of *Mg*CMP1-RNAi oysters. (**A**) The mRNA expression of *Mg*CMP1 in MF of mantle after the injection of ds*Mg*CMP1. (**B,C**) The mRNA expression levels of *Mg*collagen I and *Mg*collagen X in *Mg*CMP1-RNAi oysters. EGFP-RNAi oysters were used as control. Vertical bars represent the mean ± S.D. (N = 3). Asterisks indicate significant differences (*: $p < 0.05$, **: $p < 0.01$ and ***: $p < 0.001$).

### 3.5. Morphologic Characteristics of New-Formed Shell in Punctured Oysters under Acidification Treatment

After 2, 4 and 6 weeks of acidification treatment, the lengths of the new-formed shells were, respectively, 0.79, 0.83 and 0.84-fold shorter than those of the control group (pH 8.1 ± 0.05) (Figure 4A). The thicknesses of the new shells in the acidification treatment group (OA group, pH 7.8 ± 0.05) were 0.44, 0.37 and 0.45-fold lower than those of the control group (pH 8.1 ± 0.05), respectively (Figure 4B). These results indicated that shell formation was delayed in the acidification treatment group (OA group, pH 7.8 ± 0.05) compared to the control group (pH 8.1 ± 0.05).

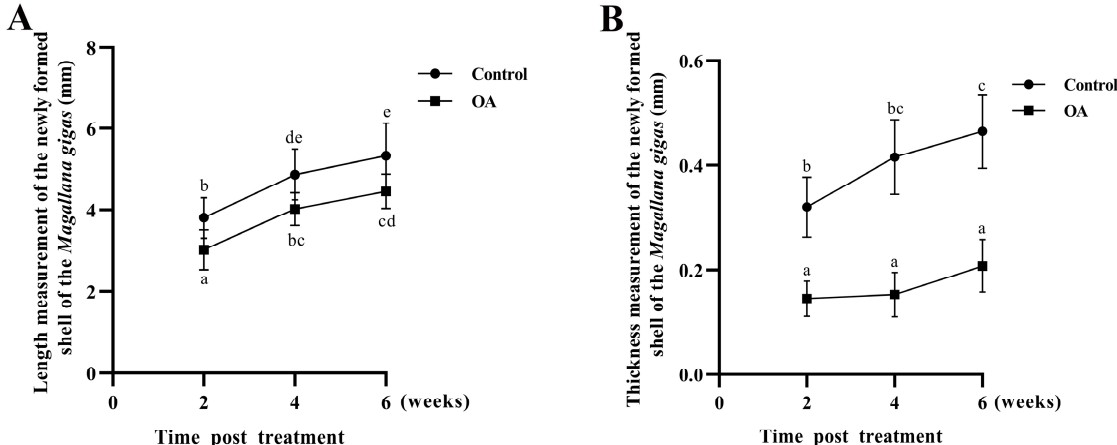

**Figure 4.** The length and thickness of new-formation oyster shells. (**A**) Length of newly formed oyster shells after perforation stimulation (control) and acidification (OA) treatment. (**B**) Thickness of newly formed oyster shells after perforation stimulation (control) and acidification (OA) treatment. Vertical bars represent the mean ± S.D. (N = 3) and the different letters (a, b, c, d and e) indicate significant differences ($p < 0.05$).

### 3.6. The mRNA Expression Levels of MgCMP1 in IF, MF and OF of Mantle of Punctured Oysters under Acidification Treatment

The expression levels of *Mg*CMP1 mRNA in the IF, MF and OF in the mantle of oysters after aperture and acidification treatment were investigated by qRT-PCR. Significant changes in *Mg*CMP1 mRNA expression were detected in MF. When the oysters were cultured at pH 8.1 for 2, 4 and 6 weeks, the expression level of *Mg*CMP1 mRNA in the MF of oysters with the aperture treatment (control group, pH 8.1 ± 0.05) was significantly increased compared to that of the oysters that had received no treatment (blank group, pH 8.10 ± 0.05) (Figure 5A–C, $p < 0.05$). After these punctured oysters were exposed to acidification treatment (OA group, pH 7.8 ± 0.05) for 2, 4 and 6 weeks, the expression level of *Mg*CMP1 mRNA in MF of was significantly decreased, 0.26-fold (Figure 5A, $p < 0.05$), 0.16-fold (Figure 5B, $p < 0.05$) and 0.19-fold (Figure 5C, $p < 0.05$), compared to those in the control group (pH 8.1 ± 0.05). No significant changes were observed in IF and OF between the control group and OA group.

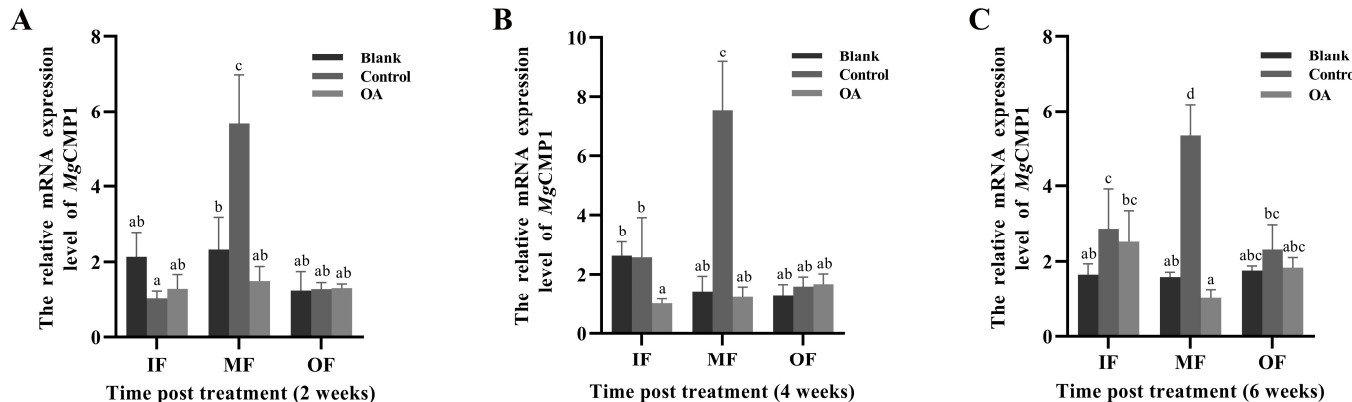

**Figure 5.** The mRNA expression level of *Mg*CMP1 under acidification treatment. (**A**) Acidification treatment 2 weeks; (**B**) acidification treatment 4 weeks; (**C**) acidification treatment 6 weeks. IF: inner fold of the mantle; MF: middle fold of the mantle; OF: outer fold of the mantle. Blank: without any treatment (blank group, pH 8.1 ± 0.05); Control: aperture treatment (control group, pH 8.1 ± 0.05); OA: acidification treatment (OA group, pH 7.8 ± 0.05). Vertical bars represent the mean ± S.D. (N = 3) and the different letters (a, b, c, and d) indicate significant differences ($p < 0.05$).

### 3.7. The Distribution of MgCMP1 mRNA Transcripts in IF, MF and OF in Mantle of Punctured Oysters under Acidification Treatment

After acidification treatment for 4 weeks, precise localization of *Mg*CMP1 mRNA transcripts in IF, MF and OF in mantle was investigated via ISH. No signals were detected in all negative groups (Figure 6A–C), and bluish-violet positive hybridization signals were distributed in the three mantle folds of the oyster (Figure 6D–F). When the oysters were cultured at pH 8.1 for 4 weeks after being punctured (control group, pH 8.1 ± 0.05), the positive hybridization signals of *Mg*CMP1 in the three mantle folds were obviously strengthened, especially in MF (Figure 6E). After these punctured oysters were exposed to acidification treatment (OA group, pH 7.8 ± 0.05), the positive hybridization signals of *Mg*CMP1 in the three mantle folds were all weakened (black arrows), especially in MF (Figure 6F).

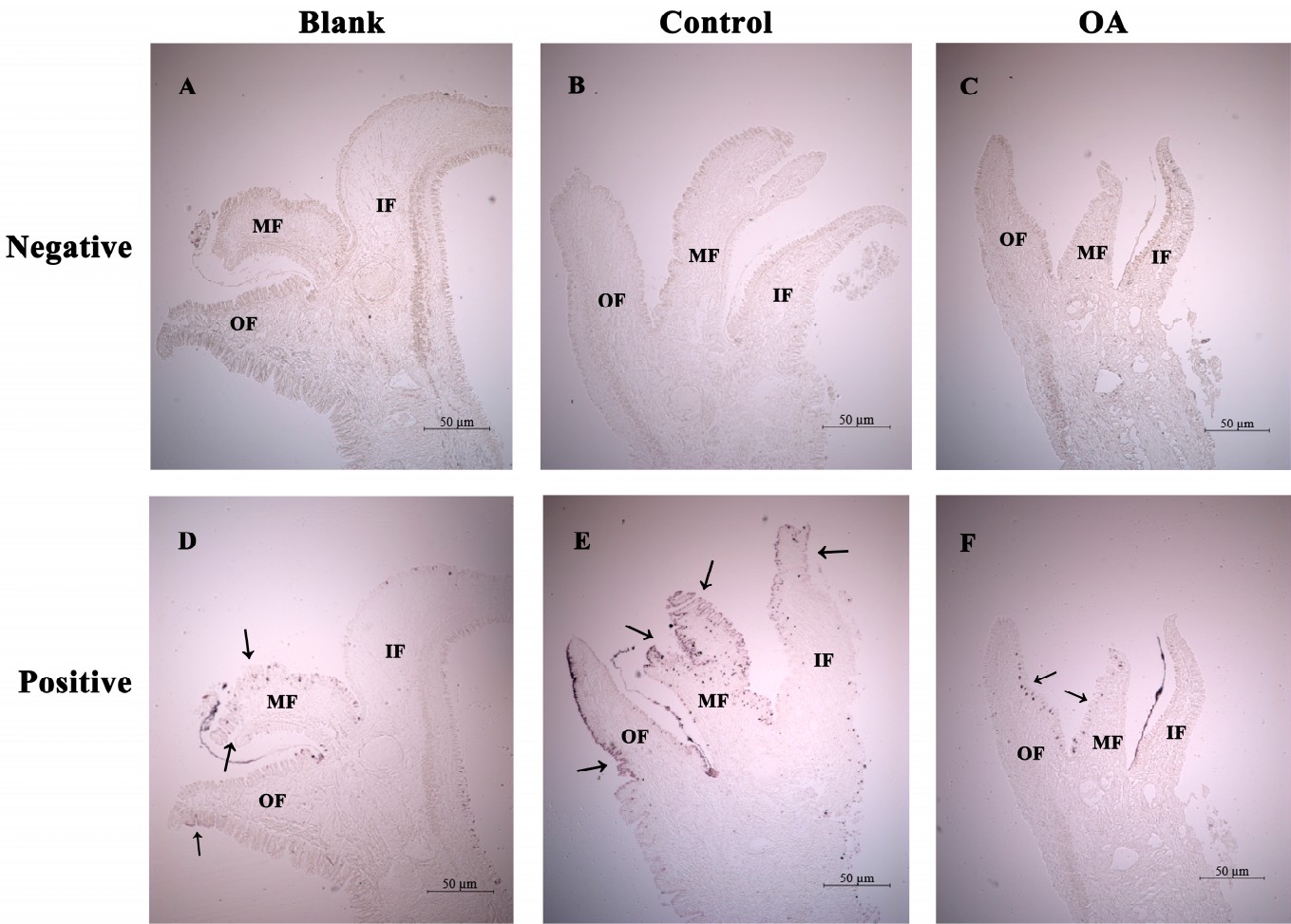

**Figure 6.** The distribution of *Mg*CMP1 mRNA transcripts in the mantle of punctured oysters under acidification treatment via ISH. (**A–C**) Righteous RNA probe hybridization (negative control); (**D–F**) antisense RNA probe hybridization (positive group). IF: inner fold of the mantle; MF: middle fold of the mantle; OF: outer fold of the mantle. Blank: without any treatment (blank group, pH 8.1 ± 0.05); Control: aperture treatment (control group, pH 8.1 ± 0.05); OA: acidification treatment (OA group, pH 7.8 ± 0.05). The positive bluish-violet signals are labeled with arrows.

*3.8. The mRNA Expression Levels of Mgcollagen I and Mgcollagen X in IF, MF and OF in Mantle of Punctured Oysters under Acidification Treatment*

The expression levels of *Mg*collagen I and *Mg*collagen X mRNA in IF, MF and OF in mantle of punctured oysters under acidification treatment for 4 weeks were investigated by qRT-PCR. Significant changes in *Mg*collagen I and *Mg*collagen X transcripts were detected in MF. The expression levels of *Mg*collagen I and *Mg*collagen X mRNA in MF of the punctured oysters in the control group (pH 8.1 ± 0.05) were significantly higher than those of the unpunctured oysters in the blank group (pH 8.1 ± 0.05) (Figure 7A,B, $p < 0.01$). After these punctured oysters were exposed to acidification treatment (OA group, pH 7.8 ± 0.05) the mRNA expression levels of *Mg*collagen I and *Mg*collagen X in MF were all significantly decreased, 0.07-fold (Figure 7A, $p < 0.01$) and 0.14-fold (Figure 7B, $p < 0.01$), compared to the control group (pH 8.1 ± 0.05).

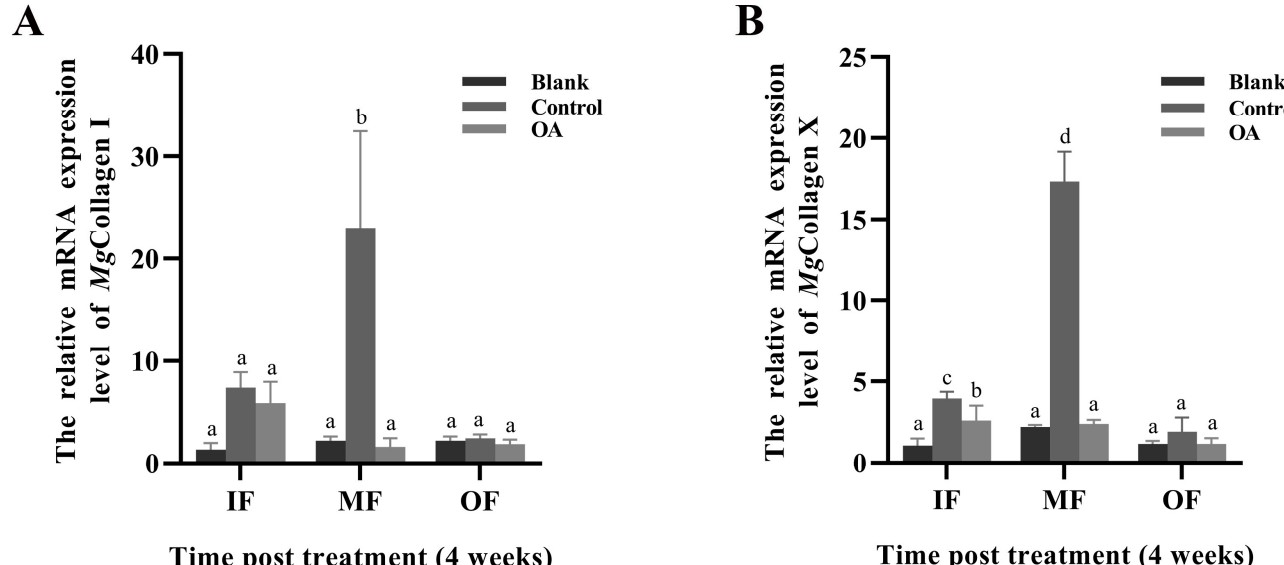

**Figure 7.** The mRNA expression levels of *Mg*collagen I (**A**) and *Mg*collagen X (**B**) under acidification treatment for 4 weeks. IF: inner fold of the mantle; MF: middle fold of the mantle; OF: outer fold of the mantle. Blank: without any treatment (blank group, pH 8.1 ± 0.05); Control: aperture treatment (control group, pH 8.1 ± 0.05); OA: acidification treatment (OA group, pH 7.8 ± 0.05). Vertical bars represent the mean ± S.D. (N = 3), and the different letters (a, b, c, and d) indicate significant differences ($p < 0.05$).

## 4. Discussion

Marine bivalves evolved shells to maintain their soft bodies, store inorganic ions, and protect themselves from predators and dehydration [52]. In recent years, OA has become a threat to many marine organisms and it inhibits shell synthesis [53]. Previous studies have demonstrated that shell organic matrix is responsible for nucleation, polymorphism, orientation, morphology and texture of calcium carbonate crystallites in the shell [54,55]. In the present study, a cartilage matrix protein (*Mg*CMP1) was identified from *M. gigas*. The response of *Mg*CMP1 secreted by the middle membrane of the mantle to acidification treatment was studied; this may help to better understand the regulatory effect of *Mg*CMP1 on oyster shell formation under acidification conditions.

During shell formation, matrix proteins are associated with the mineral phase and influence calcite $CaCO_3$ crystal growth [56–58]. In the present study, a cartilage matrix protein gene was identified from *M. gigas*. Compared with CMPs in vertebrates, *Mg*CMP1 contains only two VWA domains, which is similar to the structure of CMPs in other invertebrates [34,59]. The VWA domain is a common domain in the protein family and can interact with other extracellular substrates [22,23]. The phylogenetic tree suggested that *Mg*CMP1 was more similar to CMPs from invertebrates, especially bivalves (*P. vulgata* and *C. virginica*). These results suggest that *Mg*CMP1 is a member of the cartilage matrix protein family with a relatively conserved evolutionary position and might have similar functions in other invertebrates and vertebrates.

In vertebrates, CMP plays an important physiological role in the development of the osteoarticular system and in maintaining the homeostasis of connective tissue structures [24]. In invertebrates, CMP and VWA domain-containing proteins have been associated with shell formation in *P. fucata* and *Mytilus coruscus*. CMP may be involved in regulating the formation of prodissoconch II [18]. The VWA domain-containing protein (VDCP) in *M. coruscus* was found to be highly expressed in the mantle and associated with shell mineralization and shell attachment [60]. Oyster fasciclin-like protein (Flp), which contains a VWA domain, is also highly expressed in the mantle and involved in shell matrix maintenance [61,62]. In this study, the mRNA transcript of *Mg*CMP1 was expressed in all the

examined tissues of oyster, with the highest expression level in the mantle. It is speculated that *Mg*CMP1 might be related to shell formation in oysters. The mantle is an important tissue involved in the formation of mollusk shells, and different shell layers are related to different zones and folds of the mantle. In *P. fucata*, the nacreous layer is formed by sub-marginal zone and central zone secretions [14,63]. Periostracum protein (PPP-10) secreted form the OF of the mantle is involved in periostracum formation in *P. fucata* [13]. BMP3 secreted from MF might participate in calcium ion metabolism regulation in *P. fucata* [64]. In *M. gigas*, previous research proved that genes secreted by the MF of the mantle are involved in shell formation. Calmodulin (CaM) and engrailed, which are involved in the regulation of calcium homeostasis and the biosynthesis of periostracum, are highly expressed in the MF of mantle [17,48]. In this study, *Mg*CMP1 was highly expressed in MF of the mantle, and the expression level of *Mg*CMP1 in the MF of mantle was significantly increased after aperture treatment. This indicates that *Mg*CMP1 secreted in the MF of mantle might be involved in oyster shell formation.

It has been found that the VWA domain in CMP has a function in exercising protein–protein interactions. In vertebrates, CMPs can bind to BMP to participate in the regulation of corresponding downstream signaling pathways, affecting collagen gene expression and the integrity of the cartilage growth matrix [65,66]. In this study, interaction was identified between rMgCMP1 and rMgBMP7, indicating the possible presence of a CMP1-BMP7 signaling pathway in oysters. In addition, after inhibition of the *Mg*CMP1 gene, the mRNA expression levels of downstream *Mg*collagen I and *Mg*collagen X decreased significantly, which is similar to the results in zebrafish [32]. These results suggest that the CMP1-BMP7 pathway could regulate collagen genes, which may affect shell formation in oysters.

There are two main hypotheses for how OA affects shell formation in shellfish. On the one hand, OA reduces the saturation state of calcium carbonate minerals and affects the integrity of shell deposits [9,67–69]. On the other hand, the secretion of shell matrix proteins is inhibited by OA. In *P. fucata*, the nacreous layer formation protein (n16) and nacrein are suppressed under acidification, and then shell formation is delayed [70]. Acidification could also inhibit the expression of CaM and engrailed in the MF of mantle, affecting calcium homeostasis and shell periostracum formation in *M. gigas* [17,48]. In the present study, the growth rate of the new-formation oyster shell after acidification treatment (pH 7.8 $\pm$ 0.05) was slower than those in the control group (pH 8.1 $\pm$ 0.05), which indicates that acidification inhibited shell formation. After acidification treatment (pH 7.8 $\pm$ 0.05), the mRNA expression level of *Mg*CMP1 in the MF the of mantle was significantly reduced. The ISH result also showed that the positive *Mg*CMP1 signals in the MF of mantle were weakened. In addition, after acidification (pH 7.8 $\pm$ 0.05), the mRNA expression levels of *Mg*collagen I and *Mg*collagen X in the MF of mantle were significantly reduced. These results indicate that the activation of the CMP1-BMP7 pathway might be inhibited under acidification, which further affects collagen synthesis as well as the shell growth of oysters.

## 5. Conclusions

In summary, a homolog of CMP, *Mg*CMP1 was identified from *M. gigas*. The mRNA of *Mg*CMP1 was highly expressed in the mantle, especially in the MF of the mantle. rMgCMP1 can interact with rMgBMP7 in vitro. The mRNA expression level of *Mg*collagen I and *Mg*collagen X in the MF significantly decreased after the expression of *Mg*CMP1 was inhibited. When oysters were subjected to $CO_2$-induced acidification stress, calcified shell growth was severely inhibited, and the expression of *Mg*CMP1, *Mg*collagen I and *Mg*collagen X was significantly suppressed. The above results indicate that the CMP1-BMP7 pathway is involved in regulating the expression of *Mg*collagen I and *Mg*collagen X in the MF of the mantle in response to OA.

**Supplementary Materials:** The following are available online at https://www.mdpi.com/article/10.3390/fishes8060290/s1, Figure S1: The length and thickness of new formation oyster shells. (A) Length (labeled with red arrows) of Newly formed oyster shells after perforation stimulation and acidification treatment; (B) Thickness (labeled with yellow arrows) of Newly formed oyster shells after perforation stimulation and acidification treatment. Figure S2: The sequence characteristics of *Mg*CMP1. (A) The nucleotide and deduced amino acid sequence of *Mg*CMP1. The VWA domains are marked with a red font. (B) The structure of *Mg*CMP1 predicted by SMART. (C) Phylogenetic analysis of *Mg*CMP1 with other CMPs from different species. The *Mg*CMP1 is marked with a triangle. (D) Multiple sequence alignment of *Mg*CMP1 with CMPs from other species, including *Homo sapiens* (NP_002370.1), *Mus musculus* (NP_034899.2), *Danio rerio* (NP_001093210.1), *Cyanistes caeruleus* (XP_023796800.1), *Fulmarus glacialis* (XP_009573578.1), *Patella vulgata* (XP_050403925.1), *Mercenaria mercenaria* (XP_045215666.1), *Magallana gigas* (XP_011457035.2), *Gigantopelta aegis* (XP_041375252.1), *Crassostrea virginica* (XP_022329001.1). *Mg*CMP1 is marked with a red font.

**Author Contributions:** T.Z., C.L. and Z.L. conceived and designed the experiments; T.Z., Y.G. and X.X. performed the experiments; T.Z. and C.L. analyzed the data; T.Z. wrote the original manuscript; C.L., L.W. and L.S. revised and approved the manuscript. All authors have read and agreed to the published version of the manuscript.

**Funding:** We are grateful to all the laboratory members for their technical advice and helpful discussions. This research was supported by NSFC grant (41961124009, 32072946), National Key R & D Program (2018YFD0900606), the Distinguished Professor of Liaoning (XLYC1902012) fund from Liaoning Department of Education (LJKMZ20221123), the innovation team of Aquaculture Environment Safety from Liaoning Province (LT202009) and Dalian High Level Talent Innovation Support Program (2022RG14).

**Institutional Review Board Statement:** The study was approved by the Institutional Animal Care and Use Committee (IACUS) of Shanghai Ocean University (approval number SHOU-23-014) in February, 2023.

**Data Availability Statement:** The raw data supporting the conclusions of this article will be made available by the authors, without undue reservation.

**Acknowledgments:** We are grateful to all the laboratory members for their technical advice and helpful discussions.

**Conflicts of Interest:** The authors declare that the research was conducted in the absence of any commercial or financial relationships that could be construed as a potential conflict of interest.

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
