# Peer review of "A Cartilage Matrix Protein Regulates Collagen Synthesis in Mantle of Magallana gigas (Crassostrea gigas) under Ocean Acidification"

_fishes, doi:10.3390/fishes8060290_

Round 1
Reviewer 1 Report
Review report for the manuscript “A cartilage matrix protein regulates the collagen synthesis in mantles of Crassostrea gigas under ocean acidification”.
The study investigated the roles of cartilage matrix proteins (CMPs) in shell formation of C. gigas under the ocean acidification scenario. Overall, the MS is well-written and extremely interesting to read. I would recommend this MS for publication with few minor corrections.
Line 19: two VWA domains: full term “von Willebrand factor domains” should be appeared before abbreviation VWA.
Line 22: Full term of BMP7 is needed.
Line 48: Again, VWA is appeared before it is given full-term in the line 51.
Line 56: BMP2, BMP4 and BMP7
Line 80: Correct the typo error “Hhowever” by “However”.
Line 93: Correct the typo error “Eight-one oysters” by “Eighty-one”.
Line 111: Correct the “Total RNA rxtraction” by “Total RNA extraction”.
Lines 194-197 and lines 199-202: Repeat interpretation. I suggest deleting lines 194-197.
Fig 2B: Please correct “CgCMP2” by “CgCMP1” in the y axis.
Line 247-248: Delete “After 2, 4 and 6 weeks of drilling and acidification treatment, the new formed shell around the aperture was continuously measured using vernier calipers” as this statement was already mentioned in the methodology.
Line 380: P. fucata should be italicized.
no
Reviewer 2 Report
This paper addresses the impact of ocean acidification on Pacific oyster shell biosynthesis, especially on the role of cartilaginous matrix proteins. Overall, the article is well presented and provides sufficient data to allow a thorough evaluation of the changes promoted by the experimental design.
However, there are some comments that need to be addressed by the authors:
The authors should add information in the introduction about the VWA, what it is and what they have to choose that target.
Also, some more information is needed in the material and methods regarding environmental conditions and water quality, temperature, tank size, flow rate, among others.
In addition, a detailed explanation of how the opening treatment was done is needed.
There are some spelling mistakes in lines 80, 111 and 178. I suggest reviewing the whole document to correct these errors.
Overall, there are no experimental mistakes and the article provides interesting new data that will increase basic scientific knowledge. Therefore, I will have no problem in recommending this article for publication.
Reviewer 3 Report
The study "A cartilage matrix protein regulates the collagen synthesis in mantle of Crassostrea gigas under ocean acidification" assessed the impact of ocean acidification on shell formation in oysters assessing collagen I and X expression regulation by CgCMP1 and exploring the mechanisms inhibiting oyster shell formation. While the results seem interesting and much work was involved in this study, the paper leaves something to be desired. Not only is the work structure somewhat confusing, but it doesn't seem to have a guiding line throughout it. I would suggest a thorough review of the paper, not forgetting that not all readers have a strong background in genetics and better descriptions of the concepts and the links between them. In the introduction, the ocean acidification impacts description on bivalves should be more in-depth, and the connection with the oyster's shell and mantle better explained. Also, try to avoid introducing acronyms that will not be used again, and always define them when they are used for the first time. Please don't assume the reader knows what they mean.
Regarding the methodology, as you will see in the PDF, I believe that perhaps reorganising its structure may facilitate the understanding of the paper. Also, ensure a brief description of the methods used is added. Even if only the name of the method so that the reader can associate the results with the method. In the manuscript, as it is, the reader cannot, without consulting the bibliography, know which technique was used to obtain the data. For example, you talk about SDS-PAGE and BLI in the results, but these methods were never mentioned in the methodology section. If possible, link the various techniques used to facilitate the paper understanding. Example "We use technique X to obtain the result/compound/information Y which will then be used in technique Z."
As for the Results, this section should be revised. There are repetitions in the data descriptions, Images that perhaps should be moved for the supplementary material and some doubts regarding the number of samples used for the graphs and the statistics.
Also, the Discussion should be adressed as it seems there is much repetition of the description of the results in this section. Avoid referencing the figures in the Discussion. I believe that if you identify and address the individual objectives in the Discussion, this section will be easier to follow and understand. And your ideas will be more clear.

Round 2
Reviewer 3 Report
The paper was significantly improved; however, some points are still missing. The species name should be changed to the officially accepted one. See the PDF. Also some details form the methodology are missing.
Author Response
Please see the attachment。
